# Dissecting the Regulatory Network of Maize Phase Change in *ZmEPC1* Mutant by Transcriptome Analysis

**DOI:** 10.3390/genes13101713

**Published:** 2022-09-24

**Authors:** Xiaoqi Li, Weiya Li, Na Li, Runmiao Tian, Feiyan Qi, Juan Meng, Yajuan Jiang, Chenhui Wang, Yongqiang Chen, Zhanyong Guo, Jihua Tang, Zhanhui Zhang

**Affiliations:** 1National Key Laboratory of Wheat and Maize Crop Science, College of Agronomy, Henan Agricultural University, Zhengzhou 450046, China; 2The Shennong Laboratory, Zhengzhou 450002, China

**Keywords:** maize (*Zea mays* L.), regulatory network, phase change, mutant, RNA-seq

## Abstract

The developmental phase changes of maize are closely associated with the life span, environmental adaption, plant height, and disease resistance of the plant and eventually determines the grain yield and quality of maize. A natural mutant, *Early Phase Change 1* (*ZmEPC1*), was selected from the inbred line KN5585. Compared with the wild type plant, the *ZmEPC1* mutant exhibits deceased plant stature, accelerated developmental stages, and decreased leaf size. Through the transcriptome sequencing analysis of leaf samples at flowering stage, a total of 4583 differentially expressed genes (DEGs) were screened between the mutant and wild type, including 2914 down-regulated genes and 1669 up-regulated genes. The GO enrichment and KEGG enrichment analysis revealed that the DEGs were mainly involved in hormone response, hormone signal transduction, autophagy, JA response and signal response, photosynthesis, biotic/abiotic stress, and circadian rhythms. The RT-qPCR results revealed that the most tested DEGs display consistent expression alterations between V5 and FT stages. However, several genes showed opposite expression alterations. Strikingly, most of the JA biosynthesis and signaling pathway-related genes displayed diametrically expression alterations between V5 and FT stages. miR156, a key regulator of plant phase transition, exhibited significant down-regulated expression at V5 and FT stages. The expression of two miR156 target genes were both significantly different between mutants and wild type. In conclusion, *ZmEPC1* was identified to be mainly involved in the regulation of JA-mediated signaling pathways and hormone response and signaling, which is possible to confer developmental phase change through miR156-*SPLs* pathway.

## 1. Introduction

Maize is one of the most popular grain crops for human food, animal feed, and industrial materials. Plant growth and development largely determine plant height, yield, quality, and disease resistance in maize [1]. The development process of maize includes two developmental phase changes, juvenile to adult vegetative phase and vegetative to reproductive phase, with significant phenotypic alterations. In maize, the juvenile stage is usually from germination to five or six leaves old in most genotypes, and the adult vegetative stage is form the end of juvenile vegetative stage to flowering time [2]. The juvenile and adult leaves are distinguished primarily by features of the epidermis of the leaf blade, the most obvious of which are the presence of epicuticular wax on the juvenile leaf and epidermal hairs on the adult leaf. Flowering represents the onset of reproductive phase that mainly displays ear development, grain development, and leaf senescence. The two developmental phase transition play a crucial regulatory role in maize environmental adaption, development, and yield, which provides breeders the opportunities for selecting different types of maize varieties through manipulating the developmental phase transition timing.

The increasing studies in model plants and maize revealed that the juvenile to adult vegetative phase transition is tightly regulated by the opposite actions of two miRNAs, miR156 and miR172 [3,4,5,6,7,8]. In *Arabidopsis*, vegetative leaves are classified as either juvenile leaves or adult leaves based on their specific traits, such as leaf shape and the presence of abaxial trichomes. Juvenile leaves are characterized by high levels of miR156/157, and adult leaves are characterized by high levels of miR156/157 targets that encode SQUAMOSA PROMOTER BINDING PROTEIN-LIKE (SPL) transcription factors [9,10]. During vegetative phase change, the expression of miR156 alteration in leaf composition alters in photosynthetic traits [7]. In the target genes of miR156/157, the expression of *SPL9*, *SPL13*, and *SPL15* were found to strongly promote vegetative phase change [11]. Additionally, miRNA biogenesis related genes [9], *ABA INSENSITIVE5* (*ABI5*) [12], *VIVIPAROUS/ABI3-LIKE* (*VAL*) [13], and *BRASSINAZOLE-RESISTANT 1* (*BZR1*) [14] have been proved to regulate vegetative phase change through miR156-dependent pathway. The expression of miR172b is directly regulated by *SPL9*, and miR172 represses the expression of members of the APETALA2 (AP2) and AP2-like gene family, such as *TARGET OF EAT1* (*TOE1*), *TOE2*, *TOE3*, *SCHLAFMUTZE* (*SMZ*), and *SCHNARCHZAPFEN* (*SNZ*) [5]. Theses AP2 and AP2-like transcription factors are known to act as repressors of vegetative phase change [6,9]. Different factors, including embryonic regulators, sugar, meristem regulators, hormones, and epigenetic proteins have been proved to be involved in controlling the juvenile-to-adult vegetative phase change [9,15]. The hormones affect vegetative phase change are gibberellic acid (GA) [16], jasmonic acid (JA) [17], abscisic acid (ABA) [12], brassinosteroid (BR) [14], and auxin [9]. In maize, genetic studies have identified *teopod1* (*Tp1*), *teopod2* (*Tp2*), *teopod3* (*Tp3*), *gloosy15* (*gl15*), and *Corngrass1* (*Cg1*) in control of vegetative phase change [3,4,18]. Two hormones, GA and JA, act as critical determinants in maize juvenile to adult vegetative phase transition [16,18].

In plants, flowering is an important developmental stage that is dynamically regulated by various endogenous and environmental cues [19,20]. Signaling pathways, including photoperiod pathway and circadian clock, vernalization and autonomous pathways, GA pathway, ambient temperature pathway, age pathway, meristem responses, have been proven to be important determinants in floral induction [19,21,22,23,24,25,26]. The photoperiod and vernalization pathways control flowering in response to seasonal changes in day length and temperature; the ambient temperature pathway responds to daily growth temperatures; and the age, autonomous, and gibberellin pathways act more independently of environmental stimuli [19]. *FLOWERING LOCUS T* (*FT*) and *SUPPRESSOR OF OVEREXPRESSION OF CONSTANS 1* (*SOC1*) act as integrators for the six signaling pathways in the regulatory network of flowering time [27,28,29]. The vernalization pathway activates flowering by silencing FLC in response to prolonged exposure to low temperatures [30,31]. The photoperiod pathway acts in the leaves through a signaling cascade involving *GIGANTEA* (*GI*) and *CONSTANS* (*CO*) [24,26]. The circadian clock comprises three interlocked feed-back loops that involving the partially redundant transcription factors CIRCADIAN CLOCK ASSOCIATED 1 (CCA1) and LATE ELONGATED HYPOCOTYL (LHY) [25]. As the central repressors of the GA signaling pathway, DELLAs have been shown to physically interact with and regulate the activity of many transcription factors in leaves and shoot apices to regulate flowering [22]. The MADS box transcription factor SHORT VEGETATIVE PHASE (SVP) appears to play a crucial role in ambient temperature pathway to regulate flowering [10,24]. The age pathway is controlled by miR156-*SPLs*, which ensures plants flower under non-inductive conditions [11,32,33]. In the shoot apical meristem, TFL1 (TERMINAL FLOWER 1), a mobile protein ensures the change from vegetative to floral meristems [26]. Besides GA, other plant hormones also play important roles in the control of flowering time, including ABA, auxin, BR, JA, ethylene, and cytokinin (CTK) [6,21,22,34,35,36,37,38,39,40]. By contrast, the understanding of the genetic controls of flowering time in maize is limited. Maize exhibits tremendous natural diversity in flowering time that is controlled by a complex genetic architecture, with numerous small-effect quantitative trait loci (QTLs) involved [41,42,43]. Through map-based cloning, a series of flowering-related genes or QTLs have been identified, such as *indeterminate1* (*id1*) [44], *delayed flowering1* (*dlf1*) [40], *ZEA CENTRORADIALIS 8* (*ZCN8*) [39], *ZCN12* [45], *ZmMADS1* [46], *ZEA MAYS MADS4* (*ZMM4*) [47], *Vegetative to generative transition 1* (*Vgt1*) [48], *ZmCCT9* [49], *ZmCCT10* [50], *ZmMADS69* [51], *High Phosphatidyl Choline 1* (*HPC1*) [52], *ZmNF-YC2* [53], and *ZmCOL3* [34]. Additionally, GA and JA play opposite roles in regulating maize flowering [16,18].

In the present work, we isolated a natural mutation, *Early Phase Change 1* (*ZmEPC1*), exhibiting accelerated developmental phase changes. To identify the potential developmental phase transition related genes and construct the corresponding regulatory model, we conducted comparative transcriptome analysis between *ZmEPC1* mutant and wild type (WT) NILs.

## 2. Materials and Methods

### 2.1. Plant Materials and Growth Condition

In our previous study, an early developmental phase change mutant *ZmEPC1* was screened form the inbred line *KN5585*. The *ZmEPC1* mutant displays serious male and female imbalance, which is difficult for pollination and seed-setting. For mapping the mutant gene, we crossed the mutant with the inbred line KN5585 twice to construct a segregation population. In the constructed BC_1_F_2_ population, the dominant homozygous material was WT (almost without any phenotypic difference from KN5585), and the recessive material with early flowering and decreased plant stature was the early phase change type *ZmEPC1*. In the summer of 2020, the BC_1_F_3_ population was planted in the field and photographed during growth and development. Leaf samples (the 5th leaf of 5-leaf (V5) stage and ear leaf of flowering stage (FT)) of *ZmEPC1* mutant plants and the corresponding control were collected (3 biological replicates, respectively) and frozen in liquid nitrogen immediately. The treated samples were stored in the −80 °C freezer for further transcriptome sequencing and RT-qPCR analysis.

### 2.2. Total RNA Isolation and Transcriptome Analysis

Total RNA was extracted from the collected leaf samples of *ZmEPC1* mutants and the WT at V5 and FT stage using Trizol reagent (Invitrogen, Waltham, MA, USA) according to the manufacturer’s instructions. The RNA samples of the ear leaf samples (collected at FT stage) were used to construct 6 sequencing libraries, and the libraries were sequenced using the Illumina HiSeq 4000 platform (Berry Gene, Beijing, China). The entire original sequence data in fastq format have been uploaded to the NCBI Short Read Archive (accession number: PRJNA869324).

In order to identify the changes at the transcriptome level involved in the developmental changes of *ZmEPC1* mutant, the obtained sequencing data was analyzed. First, we performed quality control for the obtained raw data using FastQC (http://www.bioinformatics.babraham.ac.uk/projects/fastqc/, accessed on 18 August 2022). The Q30 ratios of the 6 libraries were all greater than 92%. The trimmed and low-quality (Q < 30) sequencing data were removed by Trimmomatic Software V0.39 software (http://www.usadellab.org/cms/?page=trimmomatic, accessed on 18 August 2022), 23.2 Gb sequence data was obtained. Then, the clear sequencing data were aligned to the maize B73 RefGen_V4.42 reference genome (http://ensembl.gramene.org/Zea_mays/Info/Index, accessed on 18 August 2022) by HISAT2 V2.2.1 (https://guix.gnu.org/packages/hisat2-2.2.1/, accessed on 18 August 2022). StringTie software V2.2.1 (https://github.com/gpertea/stringtie, accessed on 18 August 2022) was used to assemble the transcript and generate the count matrix. Differentially expressed genes (DEGs) were screened by the VST mode of DESeq2 software V1.30.1 (https://git.bioconductor.org/packages/DESeq2, accessed on 18 August 2022) based on |Log_2_ Fold Change| > 1 and FDR value < 0.05. The maize profile database (org. Zeamays; e.g., sqlite) was used, and the ClusterProfiler software V3.18.1 (https://git.bioconductor.org/packages/clusterProfiler, accessed on 18 August 2022) and Annotation Hub (V2.22.0, https://git.bioconductor.org/packages/AnnotationHub, accessed on 18 August 2022) R data packages were used for GO and KEGG enrichment analysis of DEGs.

### 2.3. Construction of Regulatory Network in Flowering Stage

The online tool STRING V11 (https://string-db.org/, accessed on 18 August 2022) was used to build connect network of those GO terms [54].

### 2.4. Real-Time qPCR Is Used for Gene Expression Validation

Total RNA of the leaf samples collected at the V5 and FT phases was extracted with TRIzol reagent (Invitrogen). The expression levels of miR156-*SPLs*, miR172-*gl15*, and those selected key DEGs were detected using the PrimeScript™ RT reagent kit with gDNA Eraser (Perfect Real Time) and the SYBR^®^ Premix EX Taq™ II (Tli RNaseH Plus) Kit (TaKaRa, Dalian, China). RT-qPCR primers (http://primer3.ut.ee/, accessed on 18 August 2022) are listed in Appendix A. The RT-qPCR was performed using the CFX96 Touch™ Real-Time PCR Detection System (Bio-Rad, Hercules, CA, USA). The *ACTIN* gene and U6 small RNA was used as the endogenous control for the tested genes and miRNAs, respectively. The data thus obtained were calculated by the 2^−ΔΔct^ method [55]. All experiments included 3 biological replicates and 3 technical replicates.

### 2.5. Statistical Analysis

All the collected data from RT-qPCR analysis was subjected to one-way variance analysis (ANOVA) and Student’s t-test using software SPSS 22.0 (IBM, Armonk, NY, USA). *p* < 0.05 indicates the statistical differences to reach the significant different level, *p* < 0.01 and *p* < 0.001 for very significant different level.

## 3. Results

### 3.1. Phenotypic Alterations of ZmEPC1 Mutant

The phenotypic alterations of *ZmEPC1* mutants were identified in the field (Figure 1). During the vegetative stage, the *ZmEPC1* mutant plants displayed significantly smaller plant size and more internodes than that in the WT plants (Figure 1A,B). Compared with the WT plants, the *ZmEPC1* mutant plants exhibited early tasseling (Figure 1C). The *ZmEPC1* mutant plants have significantly reduced plant height and decreased leaf size (Figure 1C). These phenotypic alterations indicated that *ZmEPC1* was involved in the regulation of maize development and the gene mutation could accelerate the developmental phase changes.

### 3.2. Identification of Differentially Expressed Genes

We obtained 23.2GB of raw data by constructing cDNA libraries and RNA-seq for 6 samples (3 replicates each for WT and *ZmEPC1* plants). Between *ZmEPC1* mutant and the WT, 4583 significantly differentially expressed genes (DEGs) were screened, including 2914 down-regulated genes and 1669 up-regulated genes. Of these DEGs, the up-regulated genes *Zm00001d051093* (encodes LRR receptor-like serine/threonine-protein kinase EFR, involved in the regulation of shoot apical meristem development), *Zm00001d039437* (encodes dbb3, involved in light signaling pathway), and *Zm00001d003811* (involved in controlling photoperiod flowering response) and down-regulated genes *Zm00001d004573* (encodes JA-inducible protein), *Zm00001d050837* (encodes gibberellin receptor-GID1L2), *Zm00001d026271* (encodes AP2/EREBP), and *Zm00001d029940* (encodes ethylene-responsive transcription factor ERF105) exhibited the most significant differences (Figure 2 and Figure 3, Table 1). These genes may play important roles in regulating maize vegetative to reproductive stage transition.

### 3.3. GO Enrichment Analysis

Based on the Annotation Hub database, gene ontology (GO) enrichment analysis was performed using the screened DEGs. The results revealed that the DEGs were mainly enriched in biological pathways, such as during photosynthesis, hormone response, cell response to hormone stimulation, cellular response to endogenous stimulation, response to endogenous stimulation, JA response, JA-mediated signal response pathway, and damage response (Figure 4).

### 3.4. KEGG Enrichment Analysis

The KEGG enrichment analysis of those DEGs indicated that the *ZmEPC1* mutation gene is mainly associated with plant hormone signal transduction, photosynthesis, linoleic acid metabolism, benzoxazinoid biosynthesis, plant-pathogen interaction, glycerophospholipid metabolism, and photosynthetic organisms. Significant biological process-related pathways were phytohormone signaling, photosynthesis, linoleic acid metabolism, and benzoxazinoid biosynthesis (Figure 5).

### 3.5. Regulatory Network Analysis

The DEGs were further analyzed to construct a biological process regulatory network involving in flowering. Photosynthesis, hormone-mediated signaling, and JA-mediated signaling are at the central places of the regulatory network (Figure 6). This indicated that the JA signaling pathway and the cellular response to JA stimulation play a crucial role in the control of maize developmental phase transition.

### 3.6. Expression Analysis of Key DEGs

In order to further verify the results of transcriptome analysis and the potential involved regulatory pathways in *ZmEPC1* mutant mediated early phase changes, 12 up-regulated genes and 18 down-regulated genes were selected for RT-PCR analysis in the samples of V5 and FT stages (Figure 7). The selected genes were mainly associated with phytohormone signaling, shoot meristem development, and photoperiod pathways. Most of the selected DEGs exhibited significantly different at V5 and FT stages. The expression trends of most genes in the V5 phase were consistent with the transcriptome results in the FT phase, which confirmed that *ZmEPC1* not only has an important regulatory role in flowering but is also involved in the regulation of vegetative phase change.

The selected up-regulated DEGs mainly involved in ethylene, GA, IAA, CTK, BR signaling pathway, as well as the photoperiod regulation pathway. Ethylene signaling pathway-related genes *Zm00001d043247* (*ETHYLENE RESPONSE SENSOR 1*) and *Zm00001d013338* (*ETHYLENE RESPONSE SENSOR 1*) were up-regulated in both V5 and FT stages. The GA signaling-related genes *Zm00001d018617* (*ga2ox12*) and *Zm00001d002999* (*ga2ox2*) were up-regulated and showed significant differences between mutants and WT at FT stage. The IAA signaling pathway-related genes *Zm00001d001945* (*arftf4*) and *Zm00001d053819* (*arftf16*) showed a down-regulated expression trend in the V5 phase and a very significant up-regulated expression trend in the FT stage. The CTK biosynthesis-related genes *Zm00001d041763* (encodes UDP-glucose) and *Zm00001d052209* (encodes glycosyltransferase) showed an up-regulated expression trend in both periods, but the difference reached a significant level only at the FT stage. The BR biosynthesis-related gene *Zm00001d033180* (*brassinosteroid-deficient dwarf1*) showed a significant down-regulation at V5 stage, while it was significantly up-regulated at the FT stage. The photoperiod regulation-related genes *Zm00001d039437* (*double B-box zinc finger protein3*) and *Zm00001d005366* (*PSEUDO-RESPONSE REGULATOR 1*) were significantly up-regulated at the V5 and FT stages.

The selected down-regulated DEGs were mainly responsible for ethylene response, GA signaling, IAA signaling, CTK signaling, BR signaling, and shoot meristem development pathways. Of these tested genes, the ethylene-responsive genes *Zm00001d049364* (*ereb209*), *Zm00001d028017* (*ereb101*), and *Zm00001d010175* (*ereb113*) were significantly down-regulated in leaves at V5 and FT stages. However, *Zm00001d029940* (encodes ethylene-responsive transcription factor ERF105) was significantly up-regulated in V5 stage and down-regulated in FT stage. The GA signaling pathway-related genes *Zm00001d050837* (encodes gibberellin receptor GID1L2), IAA biosynthesis-related genes *Zm00001d018973* (*iaa32*) and *Zm00001d018414* (*iaa24*), CTK signaling pathway-related gene *Zm00001d050371* (encodes cytokinin hydroxylase), and BR signaling pathway-related gene *Zm00001d017612* (encodes brassinosteroid-responsive RING-H2) showed extremely down-expression trend in leaves at both stages. *WOX2-Zm00001d042920* showed extremely significant down-regulation in V5 and FT phases as well.

### 3.7. Expression Analysis of Flowering Time, JA Synthesis, JA Signaling Related Genes and miR156-SPLs

Based on the results of GO and KEGG analysis, we selected the key flowering-related genes, JA biosynthesis and signaling-related genes, miR156-*SPLs* and miR172-*gl15*, for RT-qPCR analysis (Figure 8). Of the detected FT homologues, only *ZCN18* showed extremely significant up-regulation (Figure 8A). *ZCN7/8/12*, *MADS32/56/68*, and several AP2/EREBP genes all showed a significant down-regulated expression. Of the JA synthesis and JA signaling related genes (Figure 8B), only *DAD1* showed a very significant up-regulated expression, *OSAOS1*, *LOX1*, *AOC*, *Zm00001d004573*, *Zm00001d028744*, *Zm00001d048021* and other genes showed a significant down-regulated expression trend at FT stage. This suggests that *ZCN18* and *DAD1* genes may play important roles in the regulation of the early flowering of *ZmEPC1* mutant. Most of those JA biosynthesis and signaling-related genes displayed up-regulated expression at V5 stage, only *Zm00001d004573* exhibited significant down-regulated expression. Between V5 and FT stage, *OSAOS1*, *LOX1*, *AOC*, *Zm00001d028744*, and *Zm00001d048021* displayed opposite expression alterations.

The miR156-*SPLs* regulatory module has been proved to be a key regulator in plant phase transition [56]. In *ZmEPC1* mutant, the expression of miR156 was significantly reduced compared with the WT at the two detected stages (Figure 8C). The expression of the miR156 target gene *not1* was significantly increased, while *piip2* was consistent with the expression level of miR156, suggesting that *piip2* might feedback-regulated the expression of miR156. In turn, miR172 exhibited up-regulated expression in *ZmEPC1* mutant at V5 stage but displayed the down-regulated expression at FT stage (Figure 8D). The target gene of miR172, *gl15*, displayed opposite expression trends compared with the expression of miR172.

## 4. Discussion

### 4.1. ZmEPC1 Is Involved in the Regulation of Maize Developmental Phase Transition

In plants, the post-embryonic development of the shoot usually occurs in three more or less discrete temporal phases: juvenile vegetative phase, adult vegetative phase, and reproductive phase [15]. The timing of developmental phase transitions is important for plant growth, environmental adaptation, and crop production. In maize, an early phase change mutant displayed reduced juvenile vegetative phase, early flowering, and decreased plant height and leaf size [2]. Several early flowering-related mutants, such as *ZmCCT9-KO* [49] and *ZmMADS69-OE* [51], have also been identified to exhibit decreased plant height and leaf size and the late flowering mutants *gl15* [3], *ZmCOL3-OE* [34], *dlf1* [40], *id1* [44] and *ZmCCT10-OE* [50] exhibit increased plant height and leaf size. This research indicated that the developmental phase transition is tightly associated with the plant height and leaf size. In this study, *ZmEPC1* mutant plants exhibited obviously phenotypic changes, including early developmental phase transition, decreased plant height, and small leaves. These phenotypic changes revealed *ZmEPC1* to be an important regulator in maize juvenile to adult vegetative phase transition and vegetative to reproductive phase transition.

### 4.2. ZmEPC1 Acts on Phytohormones Signaling Pathway

Phytohormones, auxin, GA, CTK, ethylene, ABA, and BR, have been proved to act as crucial regulators in plant development and response to various environmental stimulus, including drought, heat, salinity stress, chilling damage, and heavy metal toxicity [17,57]. In control of plant vegetative phase transition and flowering, the regulatory roles of GA, JA, ABA, BR, auxin, ethylene, and CTK have been explored [6,9,12,14,16,17,21,22,34,35,36,37,38,39,40]. Especially, GA and JA have been defined to affect maize vegetative phase change and flowering [16,18]. In the regulation of flowering, GA signaling pathway acts a crucial determinant not only through its interaction with other endogenous signaling pathways and environmental stimulus but also via its crosstalk with other phytohormones [22]. DELLA proteins has been proved to link the GA signaling pathway with other phytohormone signaling pathways, such as JA, CTK, ABA, auxin, ethylene, and BR. The JA signaling pathway regulates flowering via controlling floral induction and its crosstalk with the GA signaling pathway [37]. In maize, GA promotes vegetative phase transition and flowering, but JA acts the opposite role in vegetative phase transition [16,18]. In the present study, numerous DEGs were screened in the transcriptome analysis of *ZmEPC1* mutants. The GO, KEGG and regulatory network analysis of these DEGs revealed that *ZmEPC1* is mainly involved in the regulation of biological pathways, including photosynthesis, hormone response, hormone-stimulated cell response, endogenous stimulation cell response, endogenous stimulation response, JA response, and JA-mediated signal response pathway. We analyzed the expression of JA biosynthesis- and signaling-genes in V5 and FT leaf samples, which demonstrated that most genes express significant down-regulation in mutant *ZmEPC1* at the FT stage but up-regulated expression at the V5 stage. The down-regulated expression of JA biosynthesis- and signaling-related genes at FT stage may contribute the early flowering, these genes down-regulated expression at the V5 stage are possible to result in narrow and short leaves. GA, ethylene, IAA, CTK, and BR signaling-related DEGs were selected for RT-qPCR verification in V5 and FT samples. Of the detected genes, two GA biosynthesis-related genes displayed up-regulated expression at the FT stage, which may promote flowering and a GA receptor encoding gene exhibited down-regulated expression at the V5 and FT stages, which can possibly result in the early developmental phase transition in *ZmEPC1*. *ZmEPC1* mutation also caused the expression alterations of other phytohormone signaling-related genes, such as IAA biosynthesis- and signaling-related genes, CTK biosynthesis-related genes, BR biosynthesis- and signaling-genes, and ethylene signaling-related genes. These results indicated that *ZmEPC1* mutation acts as an essential regulator in phytohormones signaling.

### 4.3. Potential Regulatory Mechanism of ZmEPC1 in Developmental Phase Changes

Regulatory modules, miR156-*SPLs* and miR172-*AP2s*, are crucial determinants in juvenile to adult vegetative phase transition [3,4,5,6,7,8]. In addition, embryonic regulators, sugar, meristem regulators, hormones, and epigenetic modifications may affect the juvenile to adult vegetative phase transition [9,15]. In the control of the vegetative to reproductive phase transition, signaling pathways, including photoperiod and circadian clock pathways, vernalization, and autonomous pathways, the GA pathway, ambient temperature pathway, age pathway, and meristem responses, have been identified to play important roles [19,21,22,23,24,25,26]. In maize, *Tp1*, *Tp2*, *Tp3*, *gl15*, and *Cg1* have been identified to confer vegetative phase change [3,4,18] and *id1*, *dlf1*, *ZCN8*, *ZCN12*, *ZmMADS1*, *ZMM4*, *Vgt1*, *ZmCCT9*, *ZmCCT10*, *ZmMADS69*, *HPC1*, *ZmNF-YC2*, and *ZmCol3* have been proven to affect vegetative to productive phase transition [34,39,40,44,45,46,47,48,49,50,51,52]. In the present work, the expression level of miR156 in *ZmEPC1* mutant leaves at the V5 and FT stage showed a significant down-regulation trend. However, miR172 exhibited up-regulated expression in *ZmEPC1* mutant at the V5 stage but down-regulated the FT stage. One target gene *Zm00001d014794-piip2* showed a similar expression trend, the other target gene *Zm00001d049824-not1* showed an opposite expression alteration to that of miR156. The target gene of miR172, *gl15*, displayed opposite expression trends compared with the expression of miR172. These results indicated that *ZmEPC1* may regulate maize developmental stage transitions through the miR156-*SPLs* and miR172-*gl15* regulatory modules. The transcriptome analysis revealed *ZmEPC1* to be involved in GA and JA signaling pathways. In *ZmEPC1* mutants, the GA and JA signaling-related genes display differentially expression in consistent with the early flowering. Two shoot meristem development related genes, *WOX2* (*Zm00001d042920* at V5 and FT stages) and *Zm00001d051093* (encodes LRR receptor-like serine/threonine-protein kinase EFR at the FT stage) displayed significantly down- or up-regulated expression in *ZmEPC1* mutants, which indicated meristem regulators could act as important determinants in developmental phase change. Moreover, the number of flowering time-related genes displayed differential expressions between the mutant and the wild type. These genes may contribute the early developmental phase change in *ZmEPC1* mutant. Collectively, the mutation gene can possibly act as a regulator of JA and GA signaling, which mediates the expression alterations of miR156-*SPLs*, miR172-*gl15*, to further modulate shoot meristem development and to determine the developmental phase changes in the maize *ZmEPC1* mutant.

## 5. Conclusions

A natural mutant *ZmEPC1* with significantly reduced plant height and early developmental phase was screened from an inbred line. By transcriptome analysis, major early developmental phase change-related genes were identified, and the underlying regulatory pathways in the mutant were analyzed. The present work provides the necessary support for cloning the candidate gene of *ZmEPC1* and dissecting the genetic mechanism in the maize developmental phase transition.

## Figures and Tables

**Figure 1 genes-13-01713-f001:**
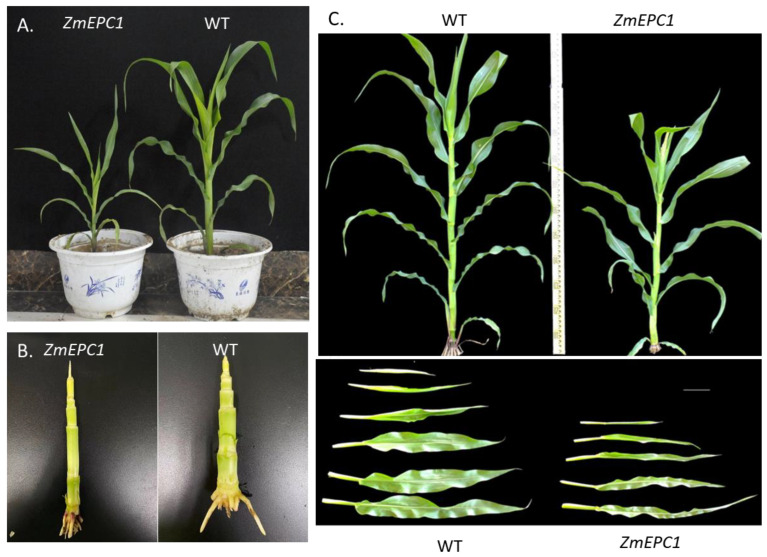
Phenotypic analysis of *ZmEPC1* mutant and the wild type. (**A**,**B**) Comparative analysis of the performance of *ZmEPC1* mutants and the wild type at vegetative phase; (**C**) Comparative analysis of the performance of *ZmEPC1* mutants and the wild type plants at tasseling stage.

**Figure 2 genes-13-01713-f002:**
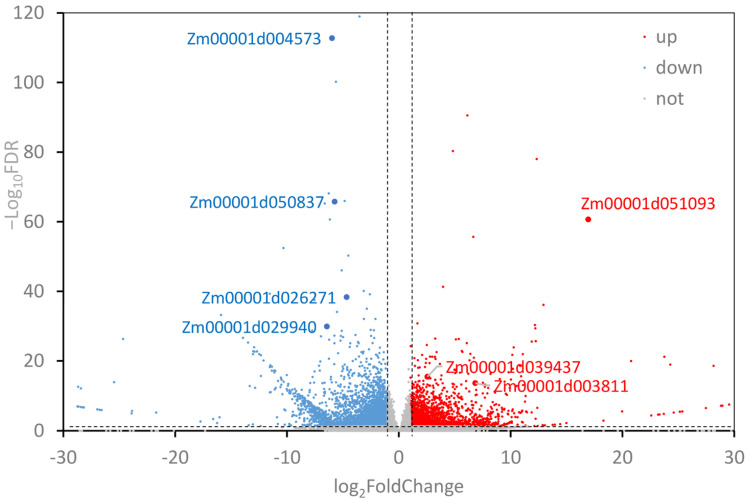
Screening of the differentially expressed genes. Red dots indicate up-regulated genes and blue dots indicate down-regulated genes (FDR < 0.05 and |Log_2_ Fold Change| > 1).

**Figure 3 genes-13-01713-f003:**
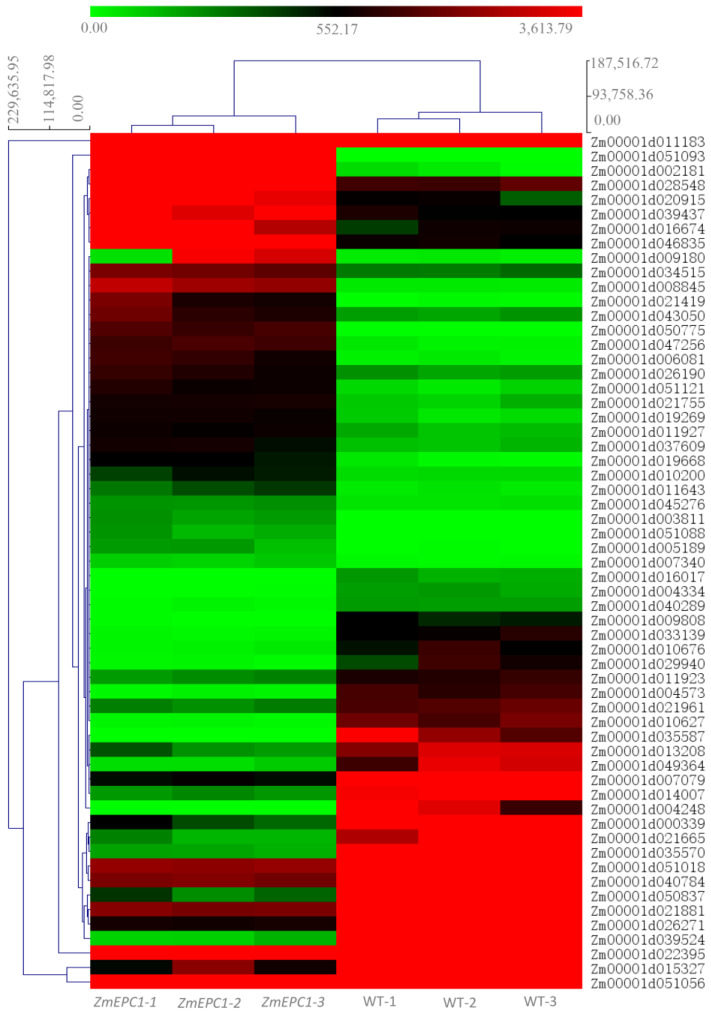
Heatmap for the screened highly significant DEGs.

**Figure 4 genes-13-01713-f004:**
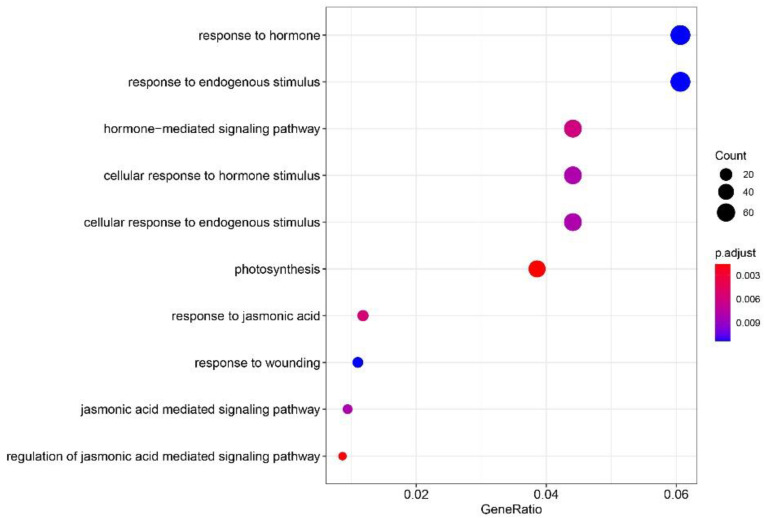
GO enrichment analysis of DEGs.

**Figure 5 genes-13-01713-f005:**
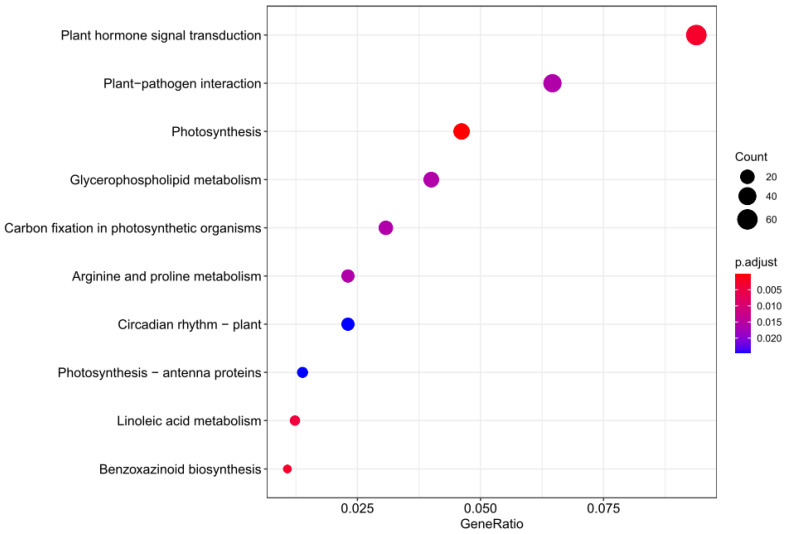
KEGG pathway enrichment analysis of DEGs.

**Figure 6 genes-13-01713-f006:**
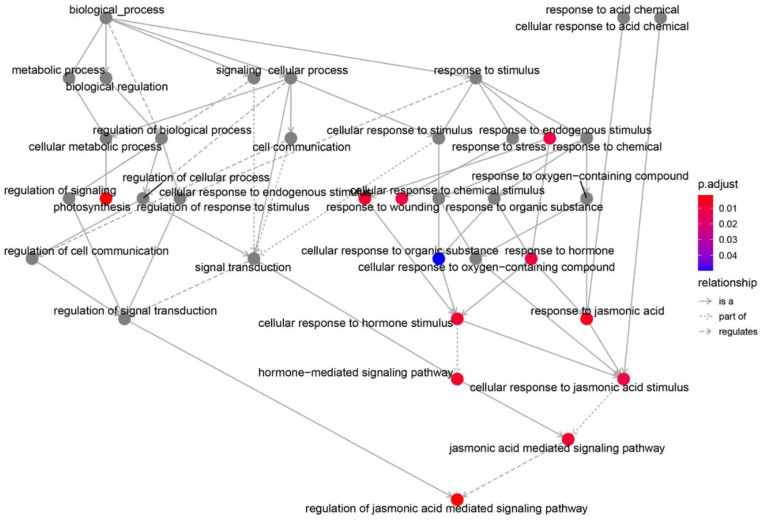
Connect network of those GO terms in biological processes. Where the relations between the GO terms are represented as edges: is a (is a subtype of); part of (part of whole); regulates (the former regulates the latter).

**Figure 7 genes-13-01713-f007:**
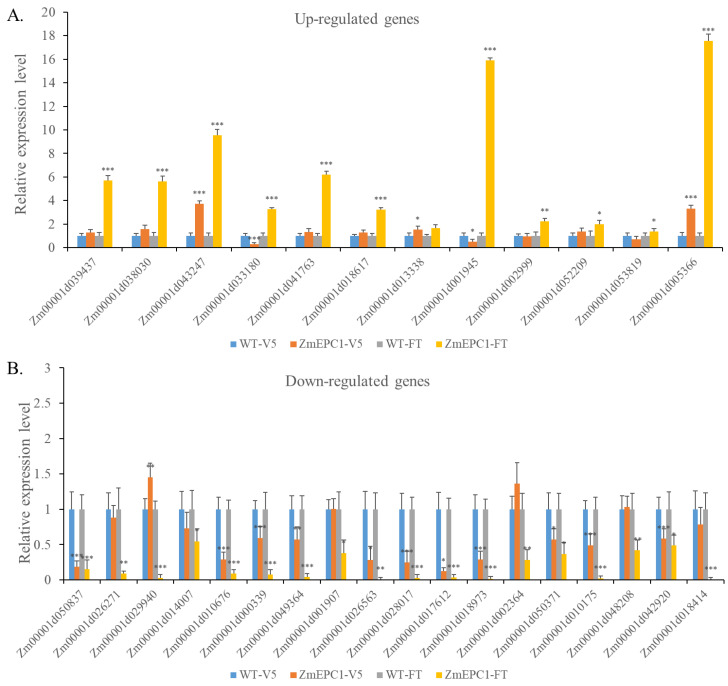
RT-qPCR analysis for the major DEGs between *ZmEPC1* and the wild type. (**A**). RT-qPCR analysis for up-regulated DEGs at V5 and FT period; (**B**). RT-qPCR analysis for down-regulated DEGs at V5 and FT period. *, **, *** represent the difference significant level at *p* < 0.05, *p* < 0.01, *p* < 0.001, respectively.

**Figure 8 genes-13-01713-f008:**
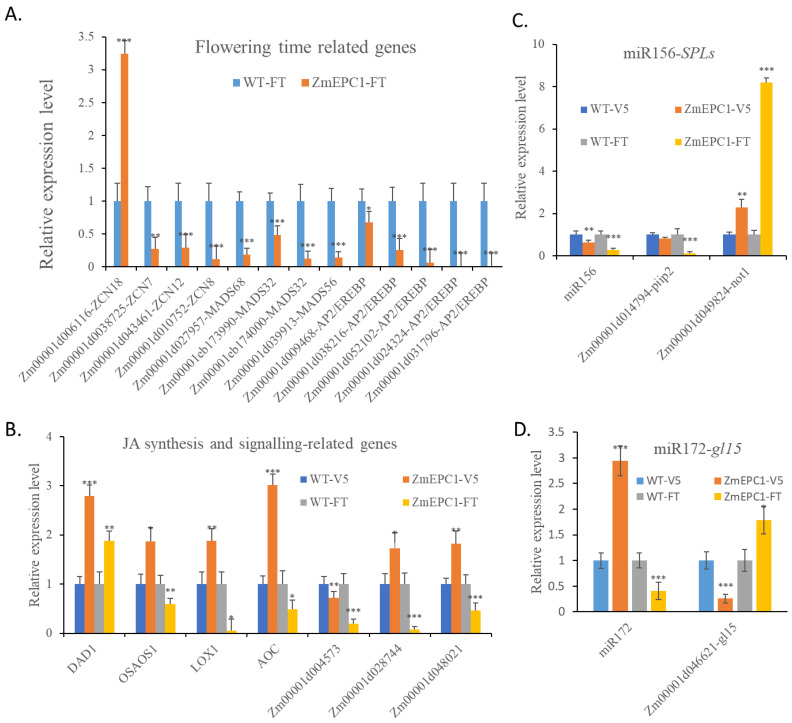
RT-qPCR for flowering stage, JA biosynthesis-related genes and miR156-*SPLs* in *ZmEPC1* mutants. (**A**) RT-qPCR analysis for flowering time related genes; (**B**) RT-qPCR analysis for JA synthesis and signaling related genes; (**C**) RT-qPCR analysis for miR156 and its target genes; (**D**) RT-qPCR analysis for miR172 and its target gene *gl15*. *, **, *** represent the difference significant level at *p* < 0.05, *p* < 0.01, *p* < 0.001, respectively.

**Table 1 genes-13-01713-t001:** Major DEGs screened by transcriptome sequencing.

Gene ID	*ZmEPC1*	WT	Log_2_ Fold Change	Padj	Gene Annotation
*Zm00001d051093*	22,414.51	0.00	16.94	2.19 × 10^−61^	LRR receptor-like serine/threonine-protein kinase EFR
*Zm00001d050775*	1391.30	0.00	12.93	7.77 × 10^−37^	NADPH-dependent pterin aldehyde reductase
*Zm00001d046835*	3674.84	649.25	2.50	2.54 × 10^−25^	rpo2; RNA polymerase2: single copy
*Zm00001d028548*	7236.83	1342.14	2.43	2.35 × 10^−21^	maternal effect embryo arrest 59
*Zm00001d051121*	804.76	83.37	3.27	2.48 × 10^−21^	RmlC-like cupins superfamily protein
*Zm00001d005189*	190.04	2.14	6.33	6.24 × 10^−21^	Cadmium/zinc-transporting ATPase HMA2
*Zm00001d021755*	821.63	122.28	2.75	4.09 × 10^−20^	UDP-glycosyltransferase 88A1
*Zm00001d037609*	712.52	140.90	2.34	5.18 × 10^−19^	GDSL esterase/lipase
*Zm00001d021419*	1240.23	4.16	8.20	3.93 × 10^−17^	Nicotinate-nucleotide pyrophosphorylase [carboxylating] chloroplastic
*Zm00001d008845*	2557.67	34.49	6.21	4.25 × 10^−17^	Pre-mRNA-processing-splicing factor 8A
*Zm00001d039437*	4271.64	727.03	2.55	2.83 × 10^−16^	dbb3; double B-box zinc finger protein3: similar to Arabidopsis light-regulated zinc finger protein 1
*Zm00001d026190*	953.45	177.84	2.42	7.27 × 10^−16^	DeSI-like protein
*Zm00001d034515*	1881.73	266.75	2.82	8.41 × 10^−16^	Homeobox-DDT domain protein RLT3
*Zm00001d011183*	112,478.99	18,093.87	2.64	9.36 × 10^−16^	thi1; thiamine biosynthesis1: low copy
*Zm00001d051088*	184.50	0.00	10.02	1.93 × 10^−15^	Putative leucine-rich repeat receptor-like protein kinase family protein
*Zm00001d011643*	371.05	38.84	3.25	2.00 × 10^−15^	Putative leucine-rich repeat receptor-like protein kinase family protein
*Zm00001d019269*	731.46	77.18	3.25	2.59 × 10^−15^	Pentatricopeptide repeat-containing protein
*Zm00001d011927*	688.27	151.24	2.19	4.50 × 10^−15^	Phospholipid/glycerol acyltransferase family protein
*Zm00001d009180*	2249.54	34.74	6.02	5.03 × 10^−15^	Folate-biopterin transporter 1 chloroplastic
*Zm00001d002181*	8050.18	43.95	7.52	1.52 × 10^−14^	G-type lectin S-receptor-like serine/threonine-protein kinase B120
*Zm00001d003811*	217.64	1.86	6.84	2.13 × 10^−14^	Two-component response regulator-like APRR1
*Zm00001d006081*	1093.69	38.92	4.81	2.80 × 10^−14^	Pleckstrin homology (PH) domain superfamily protein
*Zm00001d016674*	3864.89	662.37	2.54	5.21 × 10^−14^	Heat shock factor protein 2
*Zm00001d047256*	1355.16	28.39	5.58	5.22 × 10^−14^	Protein kinase domain superfamily protein
*Zm00001d007340*	105.04	9.27	3.51	6.62 × 10^−14^	ADP-ribosylation factor GTPase-activating protein AGD12
*Zm00001d010200*	475.70	73.10	2.70	9.45 × 10^−14^	ATP binding protein
*Zm00001d019668*	543.90	19.24	4.83	2.58 × 10^−13^	P-loop containing nucleoside triphosphate hydrolases superfamily protein
*Zm00001d043050*	1292.41	233.74	2.47	2.70 × 10^−13^	RING-H2 finger protein ATL74
*Zm00001d045276*	233.19	54.10	2.11	4.34 × 10^−13^	Lactoylglutathione lyase/glyoxalase I family protein
*Zm00001d020915*	5139.10	596.47	3.11	4.45 × 10^−13^	Pirin-like protein 2
*Zm00001d004573*	19.69	1264.00	−5.98	1.70 × 10^−113^	60 kDa jasmonate-induced protein
*Zm00001d015327*	1181.74	91,543.55	−6.28	6.72 × 10^−69^	ubi1; ubiquitin1: genomic sequence
*Zm00001d035570*	185.35	5360.00	−4.85	1.00 × 10^−66^	α/β-Hydrolases superfamily protein
*Zm00001d050837*	338.61	18,172.00	−5.75	1.63 × 10^−66^	Gibberellin receptor GID1L2
*Zm00001d010627*	18.94	1873.88	−6.62	5.24 × 10^−66^	Protein LURP-one-related 8
*Zm00001d004248*	1.81	5168.05	−11.55	4.66 × 10^−40^	UDP-glycosyltransferase 85A7
*Zm00001d021961*	265.56	1601.37	−2.59	6.62 × 10^−40^	α-L-fucosidase 3
*Zm00001d026271*	806.06	20,462.16	−4.67	4.35 × 10^−39^	Putative AP2/EREBP transcription factor superfamily protein
*Zm00001d009808*	2.72	479.37	−7.56	3.00 × 10^−38^	Probable galacturonosyltransferase 7
*Zm00001d021881*	2069.86	15,095.67	−2.87	9.70 × 10^−36^	Root border cell-specific protein
*Zm00001d033139*	17.39	803.40	−5.53	7.63 × 10^−35^	Cytochrome P450 71D7
*Zm00001d011923*	244.42	1037.29	−2.08	8.32 × 10^−33^	BTB/POZ domain-containing protein
*Zm00001d007079*	558.49	4887.17	−3.13	8.37 × 10^−33^	Phospholipase A2 family protein
*Zm00001d029940*	14.07	1216.41	−6.43	1.30 × 10^−30^	Ethylene-responsive transcription factor ERF105
*Zm00001d022395*	8199.81	47,204.65	−2.53	1.09 × 10^−29^	Rhythmically expressed protein
*Zm00001d051056*	22,281.59	105,340.39	−2.24	1.41 × 10^−29^	S-adenosylmethionine decarboxylase proenzyme
*Zm00001d039524*	121.19	26,122.74	−7.75	1.85 × 10^−29^	Transposon protein CACTA%2C En/Spm sub-class
*Zm00001d014007*	234.22	8122.34	−5.12	2.19 × 10^−29^	senescence regulator
*Zm00001d040289*	14.74	220.77	−3.88	3.27 × 10^−28^	DUF4378 domain protein
*Zm00001d021665*	195.07	10,587.20	−5.76	5.32 × 10^−28^	PRAS-rich protein
*Zm00001d010676*	30.35	1108.32	−5.19	6.00 × 10^−28^	Ethylene-responsive transcription factor 12
*Zm00001d040784*	2035.80	11,391.05	−2.48	6.40 × 10^−28^	Glycine-rich domain-containing protein 1
*Zm00001d000339*	423.35	13,472.84	−4.99	1.46 × 10^−27^	Putative AP2/EREBP transcription factor superfamily protein ereb92
*Zm00001d035587*	0.00	2416.06	−13.94	2.15 × 10^−27^	Putative S-locus receptor-like protein kinase family protein
*Zm00001d051018*	2320.48	10,694.66	−2.20	2.46 × 10^−27^	dbb4; double B-box zinc finger protein4:
*Zm00001d016017*	0.00	201.76	−24.66	4.42 × 10^−27^	Protein disulfide isomerase-like 1–2
*Zm00001d004334*	2.70	207.13	−6.23	7.24 × 10^−27^	ATG8-interacting protein 1
*Zm00001d013208*	272.99	3432.28	−3.65	3.45 × 10^−26^	Zinc finger protein AZF2
*Zm00001d049364*	85.17	4635.09	−5.77	7.57 × 10^−26^	Ethylene-responsive transcription factor

## Data Availability

Data are openly available in a public repository. The raw data of transcriptome sequencing in this study are provided at NCBI short read archive (accession number: PRJNA869324).

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
