# Peer review of "Dissecting the Regulatory Network of Maize Phase Change in ZmEPC1 Mutant by Transcriptome Analysis"

_genes, 2022, doi:10.3390/genes13101713_

Round 1
Reviewer 1 Report
The manuscript conducted transcriptome sequencing analysis between ZmEPC1 and the wild type, identified DEPs mainly involved in hormone response, hormone signal transduction, autophagy, JA response and signal response, photosynthesis, biotic/abiotic stress, and circadian rhythms, and drawed a conclusion that ZmEPC1 was mainly involved in the regulation of JA-mediated signaling pathways and hormone response and signaling to confer developmental phase change through miR156-SPLs pathway. It is meaningful to reavel the mechanism of ZmEPC1 controlling the developmental phase change. However, there are some minor item should be improved before publishment.
1. The english language need to be improved. There are some grammar errors. For examper, lines 107-111, the subject was lacked for the last two sentences. It can be corrected to "we conducted comparative transcriptome analysis be-tween ZmEPC1 mutant and wild type (WT) NILs to identify the potential developmental
phase transition related genes, to predict the regulatory model of ZmEPC1 in maize early phase transition, and to construct the corresponding regulatory
model." Line 114-115, it is indistinct for the sentence "the early developmental phase change mutant ZmEPC1 form the inbred line KN5585". My understanding is that ZmEPC1 derivated from the inbred line KN5585. Lines 115-116, It can be corrected to "The ZmEPC1 mutant displays serious male and female unbalance, which is difficult to reproduce seeds". There are too many other errors in language.
2. The figures legend in fiure 2,3 should be added to inform what the color represent. What is the mt, ck in figure 2? In figure 6, what is the meaning of isa, part of, regulators?
Author Response
Dear the Editor and Reviewers,
Thank you very much for your critical comments for our manuscript. We have revised the manuscript carefully according to your suggestions. The major revisions are as following,
1. The english language need to be improved. There are some grammar errors. For examper, lines 107-111, the subject was lacked for the last two sentences. It can be corrected to "we conducted comparative transcriptome analysis be-tween ZmEPC1 mutant and wild type (WT) NILs to identify the potential developmental phase transition related genes, to predict the regulatory model of ZmEPC1 in maize early phase transition, and to construct the corresponding regulatory model." Line 114-115, it is indistinct for the sentence "the early developmental phase change mutant ZmEPC1 form the inbred line KN5585". My understanding is that ZmEPC1 derivated from the inbred line KN5585. Lines 115-116, It can be corrected to "The ZmEPC1 mutant displays serious male and female unbalance, which is difficult to reproduce seeds". There are too many other errors in language.
We have revised these grammar errors and edited the English writing of the manuscript carefully.
2. The figures legend in fiure 2,3 should be added to inform what the color represent. What is the mt, ck in figure 2? In figure 6, what is the meaning of isa, part of, regulators?
We have redrawn figure 2 and added a figure legend. We have also revised Figure 3 and 6, as well as added figure legends.
Reviewer 2 Report
1.The English language of the article should be described more idiomatically and accurately.such as line13,14:the The description of the phenotype should be more detailed and specialized.line 24,The expression levels of miR156 exhibited significant down-regulated expression in (?stage).line 32,crop is a countable noun;line 37, the first five or six in most genotypes?;line 59,miR172b is a direct target of SPL9 ?line 62,factor→factors?line109-110,this sentence“ to predict the regulatory model of ZmEPC1 in maize early phase transition; and to construct the corresponding regulatory model” is redundant. line 275, miR156- SPLs for qRT-PCR analysis is not presented in Fig8.A/B,line 313,")";line 323, promotes promote ;line 331,ant;line 376, dissecting→dissect.Please carefully correct the grammatical errors in the article
2.Fig.1C, the tassel of maize in picture is not obvious.
3.Fig.2,It would be better if up-regulated and down-regulated genes are represented by different colors.The point of highly significant DEGs or important genes mentioned in text needed to be specific marked in Fig.2.
4.Why not detect the expression levels of miR172 target genes? the role of miR172 is introduced extensively in the introduction.
5. In 4.3 Potential Regulatory Mechanism of ZmEPC1 in Developmental Phase Changes,what are the underlying regulatory mechanisms that should be summarized in points.
Author Response
Dear the Editor and Reviewers,
Thank you very much for your critical comments for our manuscript. We have revised the manuscript carefully according to your suggestions. The major revisions are as following,
1.The English language of the article should be described more idiomatically and accurately.such as line13,14:the The description of the phenotype should be more detailed and specialized.line 24,The expression levels of miR156 exhibited significant down-regulated expression in (?stage).line 32,crop is a countable noun; line 37, the first five or six in most genotypes?;line 59,miR172b is a direct target of SPL9 ?line 62,factor→factors?line109-110,this sentence“ to predict the regulatory model of ZmEPC1 in maize early phase transition; and to construct the corresponding regulatory model” is redundant. line 275, miR156- SPLs for qRT-PCR analysis is not presented in Fig8.A/B,line 313,")";line 323, promotes promote ;line 331,ant;line 376, dissecting→dissect. Please carefully correct the grammatical errors in the article
We have revised these grammar errors and edited the English writing of the manuscript carefully.
2. Fig.1C, the tassel of maize in picture is not obvious.
This picture is improper. But we don’t have any other suitable picture to replace it. We have taken the present photos in 2020. In the next two growth seasons, however, we have not obtained any quality pictures and seeds for serious blooding and heat stress.
3. Fig.2, It would be better if up-regulated and down-regulated genes are represented by different colors. The point of highly significant DEGs or important genes mentioned in text needed to be specific marked in Fig.2.
We have remade Figure 2 and add a figure legend.
4.Why not detect the expression levels of miR172 target genes? the role of miR172 is introduced extensively in the introduction.
We have performed RT-qPCR analysis for miR172 and gl15 using the V5 and FT samples, and add the results in the manuscript.
5. In 4.3 Potential Regulatory Mechanism of ZmEPC1 in Developmental Phase Changes, what are the underlying regulatory mechanisms that should be summarized in points.
The ZmEPC1 mutation gene alters the expression of phytohormone signaling pathways (especially JA and GA signaling pathways) related genes, miR156-SPLs, miR172-gl15, flowering-related genes, and shoot meristem development related genes. Based these results, we propose that the mutation gene belong to a regulator of JA and GA signaling, which mediates the expression alterations of miR156-SPLs, miR172-gl15, further to modulate shoot meristem development and to determine the developmental phase changes in maize ZmEPC1 mutant. We have added the description in the discussion section.
Reviewer 3 Report
Li and colleagues, in this study, tried to establish the regulatory networks during maize early phase change by conducting a transcriptome analysis in the ZmEPC1 mutant ear leaves. To this end, they found 4583 differentially expressed genes (DEGs), the DEGs belong to multiple signaling pathways. They also confirmed 12 up-regulated genes and 18 down-regulated genes expression by qRT-PCR analysis. Additionally, they further detected gene expression related to JA and miR156 pathways, respectively. Their work provides novel insights into phase changes during maize development, and will provide novel knowledge for maize breeding. Generally, the manuscript is clearly written, although the study lacks genetic data considering the difficulties in genetic manipulation of maize.
Major concerns:
(1) Figure 1, the author showed the phenotypes of ZmEPC1 mutant, and some quantifications can be added such as plant height, leaf seize et al. at maize developmental stages. And please also add scale bar.
(2) As mentioned by the author, some key players have been identified in regulation of maize vegetative phase changes or vegetative to productive phase transition, however, they do not detect these key players in their transcriptome analysis. The expression level of these essential players that were not included in Figure 8A need be clarified between WT and ZmEPC1 mutant.
(3) miR172-AP2s pathway also is essential for juvenile to adult vegetative phase transition, and the gene expression of this pathway can be added from the transcriptome analysis, if possibile.
Minor concerns:
(1) Lin37, the “from germination to the first five or six leaves in most genotypes”.
(2) Line 179, Identification of differentially expressed genes.
(3) Line 327, biological pathways including photosynthesis.
Author Response
Dear the Editor and Reviewer,
Thank you very much for your critical comments for our manuscript. We have revised the manuscript carefully according to your suggestions. The major revisions are as following,
Major concerns:
(1) Figure 1, the author showed the phenotypes of ZmEPC1 mutant, and some quantifications can be added such as plant height, leaf seize et al. at maize developmental stages. And please also add scale bar.
We have only taken photos in the present field experiment. In the next two growth seasons, we have not obtained any seeds for serious blooding and heat stress. At present, we don’t have enough seeds for planting field experiment. Thus, we can’t add these data.
(2) As mentioned by the author, some key players have been identified in regulation of maize vegetative phase changes or vegetative to productive phase transition, however, they do not detect these key players in their transcriptome analysis. The expression level of these essential players that were not included in Figure 8A need be clarified between WT and ZmEPC1 mutant.
In maize, genetic studies have identified teopod1 (Tp1), teopod2 (Tp2), teopod3 (Tp3), gloosy15 (gl15) and Corngrass1 (Cg1) in control of vegetative phase change. Two hormone, GA and JA, act as critical determinants in maize juvenile to adult vegetative phase transition. Through map-based cloning, a series of flowering-related genes or QTLs have been identi-fied, such as indeterminate1 (id1), delayed flowering1 (dlf1), ZEA CENTRORADIALIS 8 (ZCN8), ZCN12, ZmMADS1, ZEA MAYS MADS4 (ZMM4), Vegetative to generative transition 1 (Vgt1), ZmCCT9, ZmCCT10, ZmMADS69, High Phosphatidyl Choline 1 (HPC1), ZmNF-YC2, and ZmCOL3. And, GA and JA play opposite roles in regulating maize flowering. Mutation genes Tp1, Tp2, andTp3 have not been cloned. We have detected the expression of miR156 (miR156 up-regulated express in Cg1 mutant), and added the expression of gl15 in the present revision. We have also detected the expression of flowering-related genes, GA signaling-related genes, and JA synthesis and signaling-related genes. These tested genes are almost involved in the possible regulatory pathways of maize vegetative phase change and vegetative to productive phase transition.
(3) miR172-AP2s pathway also is essential for juvenile to adult vegetative phase transition, and the gene expression of this pathway can be added from the transcriptome analysis, if possible.
We have performed RT-qPCR analysis for miR172 and gl15 using the V5 and FT samples, and add the results in the manuscript.
Minor concerns:
(1) Lin37, the “from germination to the first five or six leaves in most genotypes”.
We have revised this sentence.
(2) Line 179, Identification of differentially expressed genes.
We have revised this grammatical error.
(3) Line 327, biological pathways including photosynthesis.
We have revised this grammatical error.
Round 2
Reviewer 3 Report
The authors put great effort to address my comments, as some can not be addressed due to some technical issues. The manuscript has been well revised, and will povide novel knowledge to study the phase change mediated by ZmEPC1 mutation.
Minor concerns:
The Figure 8 has been inserted twice, please revised it.
Author Response
Dear the Editor and Reviewer,
Thank you very much for your critical comments for our manuscript once again. We have revised the manuscript carefully according to your suggestions.
The Figure 8 has been inserted twice, please revised it.
We have checked the figure insertions carefully, and revised this typing error.